# Effect of Flagellin Pre-Exposure on the Inflammatory and Antifungal Response of Bronchial Epithelial Cells to Fungal Pathogens

**DOI:** 10.3390/jof8121268

**Published:** 2022-11-30

**Authors:** Jeanne Bigot, Manon Ruffin, Juliette Guitard, Sandra Vellaissamy, Sophie Thorez, Harriet Corvol, Loïc Guillot, Viviane Balloy, Christophe Hennequin

**Affiliations:** 1Centre de Recherche Saint-Antoine (CRSA), Inserm, AP-HP, Hôpital Saint-Antoine, Service de Parasitologie-Mycologie, Sorbonne Université, F-75012 Paris, France; 2Centre de Recherche Saint-Antoine (CRSA), Inserm, Sorbonne Université, F-75012 Paris, France; 3AP-HP, Hôpital Saint-Antoine, Service de Parasitologie-Mycologie, F-75012 Paris, France; 4Centre de Recherche Saint-Antoine (CRSA), Inserm, AP-HP, Hôpital Trousseau, Service de Pneumologie Pédiatrique, Sorbonne Université, F-75012 Paris, France

**Keywords:** fungal infection, bronchial epithelial cells, innate immune memory, chronic pulmonary diseases

## Abstract

Bronchial epithelial cells (BEC) play a crucial role in innate immunity against inhaled fungi. Indeed, in response to microorganisms, BEC synthesize proinflammatory cytokines involved in the recruitment of neutrophils. We have recently shown that BEC exert antifungal activity against *Aspergillus fumigatus* by inhibiting filament growth. In the present study, we first analyzed the inflammatory and antifungal responses of BEC infected by several fungal species such as *Aspergillus* spp., *Scedosporium apiospermum* and *Candida albicans*, which are frequently isolated from the sputum of people with chronic pulmonary diseases. The airways of these patients, such as people with cystic fibrosis (pwCF), are mainly colonized by *P. aeruginosa* and secondary by fungal pathogens. We have previously demonstrated that BEC are capable of innate immune memory, allowing them to increase their inflammatory response against *A. fumigatus* following a previous contact with *Pseudomonas aeruginosa* flagellin. To identify the impact of bacteria exposure on BEC responses to other fungal infections, we extended the analysis of BEC innate immune memory to *Aspergillus* spp., *Scedosporium apiospermum* and *Candida albicans* infection. Our results show that BEC are able to recognize and respond to *Aspergillus* spp., *S. apiospermum* and *C. albicans* infection and that the modulation of BEC responses by pre-exposure to flagellin varies according to the fungal species encountered. Deepening our knowledge of the innate immune memory of BEC should open new therapeutic avenues to modulate the inflammatory response against polymicrobial infections observed in chronic pulmonary diseases such as CF.

## 1. Introduction

Chronic pulmonary diseases such as cystic fibrosis (CF), chronic obstructive pulmonary disease (COPD) or severe asthma are characterized by reduced mucociliary clearance. This promotes the implantation of microorganisms that are the source of recurrent infections and chronic inflammation responsible for lung parenchyma damage that could lead to end-stage respiratory failure [1]. Bacteria such as *Staphylococcus aureus* and *Pseudomonas aeruginosa* are the predominant causes of these infections [2,3]. However, fungal pathogens, notably *Aspergillus* spp., also participate in the pathophysiology of lung parenchyma damage. In the adult population, approximately 35% of people with CF (pwCF) and 13–29% of people with COPD are colonized with *Aspergillus fumigatus,* the main pathogenic *Aspergillus* species [4,5,6]. Whether bronchial colonization with *A. fumigatus* has harmful consequences by itself remains unclear, it is the starting point of different clinical forms ranging from bronchitis to allergic bronchopulmonary aspergillosis (ABPA) that undoubtedly negatively impact the respiratory performance [7,8]. In the absence of treatment, ABPA may evolve toward the development of bronchiectases and/or pulmonary fibrosis, a condition that eventually requires lung transplantation in its end stage. Furthermore, the use of immunosuppressive therapies in this particular setting exposes patients to the risk of invasive pulmonary aspergillosis [9]. Although *A. fumigatus* is the predominant filamentous fungal species isolated in these contexts, other inhaled filamentous fungi can also colonize the bronchial airways of patients with chronic pulmonary diseases. Of these, *Aspergillus flavus*, *Aspergillus terreus* and *Scedosporium apiospermum* are the most common [10,11]. On another hand, endogenous fungi, such as *Candida* spp., commensals of the digestive tract, are also frequently isolated from the bronchial airways of patients with chronic pulmonary diseases [12]. Some studies have reported that up to 93% of pwCF and 45.2% of people with COPD are colonized with *C. albicans* [13,14]. The consequences of *C. albicans* airway colonization are controversial. Some authors have shown that repeated *C. albicans* isolations are associated with an increased number of bronchitis exacerbations requiring hospitalization and a more rapid decline in lung function [15], while others claimed that *C. albicans* colonization has no deleterious effects [16].

The bronchial immune defense is mainly non-specific, represented by mucociliary clearance. Thanks to cellular differentiation, bronchial epithelial cells (BEC) secrete the mucus that traps inhaled particles and microorganisms, which are further eliminated through the ciliary beating of BEC. In addition, these cells are the first of the lower respiratory tract encountered by inhaled pathogens, and they play an important role in the innate immune response after the specific recognition of microorganisms. Indeed, BEC express highly conserved pattern recognition receptors (PRR) [17], allowing the recognition of microbial motifs called pathogen-associated molecular patterns (PAMP). The interaction between cells and microbes initiates cascades of intracellular signals leading to the activation of different mechanisms of innate immune defense. For example, it was shown that the interaction between BEC and *A. fumigatus* induces the secretion of pro-inflammatory cytokines and antimicrobial peptides (AMP) [18], the latter exhibiting antimicrobial activity against a variety of pathogens including fungi [19]. Moreover, BEC limit the filament formation of *A. fumigatus* through a PI3 kinase activation after the recognition of the FleA lectin of the fungus [20]. Otherwise, there is a growing body of evidence showing that exposure of cells of the innate immune system to microbial components can modify their responses to subsequent triggers [21,22,23]. This phenomenon, called innate immune memory (IIM), is characterized by a suppressed (tolerance) or enhanced (training) immune response to a secondary microbial stimulus. First described in monocytes [21], the IIM concept has been extended to other innate immune cells types, such as natural killer cells [24], hematopoietic progenitor cells [25], skin epithelial cells [26] and more recently, BEC [27]. Indeed, in a recent proof-of-concept study, we demonstrated that BEC first exposed to *P. aeruginosa* flagellin are able to enhance their response to a subsequent infection with *A. fumigatus*.

The aim of the present study was first, to investigate the inflammatory and antifungal responses of BEC against a panel of pathogenic fungi commonly isolated from the bronchial airways of patients with chronic pulmonary disease. Then, to analyze the role of IIM in the possible regulation of these responses.

## 2. Materials and Methods

### 2.1. Strains

Fourteen clinical strains representing the most commonly isolated species from the bronchial airways of patients with chronic pulmonary diseases were used for this study: *A. fumigatus* (*n* = 3), *A. flavus* (*n* = 4), *A. terreus* (*n* = 2), *S. apiospermum* (*n* = 2) and *C. albicans* (*n* = 3). The identification of all strains was confirmed at the cryptic species level using MALDI-TOF mass spectrometry analysis and the MSI database (https://msi.happy-dev.fr/identification/new/, accessed on: 20 January 2019). The laboratory reference strain of *A. fumigatus*, DAL (CBS 144.89), used for our proof-of-concept study [27], was added to the analysis.

Filamentous fungi were grown at room temperature onto Sabouraud dextrose agar containing chloramphenicol and gentamicin (Bio–Rad, Marnes-la-Coquette, France). After a 7-day incubation period, spores were recovered in 1 mL of sterile 0.1% PBS Tween 20. The resulting suspension was then filtered through a cell filter (40 µm Ø) and centrifuged for 10 min at 3000× *g*. The pellet was resuspended in 500 µL of 0.1% Tween 20 PBS, and the inoculum was then adjusted in an antibiotic-free culture medium to obtain the desired concentrations defined in the results section. *C. albicans* strains were subcultured onto Chromagar medium plates (Becton-Dickinson, Pont de Claix, France) and incubated at 37 °C for 2 days. The inoculum was prepared by adapting the protocol described by Reihill et al. [28]. Three colonies were inoculated into yeast peptone dextrose (YPD) broth and incubated for 7 h at 37 °C under orbital agitation at 100 rpm to reach the exponential growth phase. After centrifugation, the pellet was washed, resuspended in 1 mL sterile PBS and adjusted to the desired concentration in an antibiotic-free culture medium.

### 2.2. Cell Culture and In Vitro Model

The protocol was previously described by Bigot et al. [27]. Briefly, human bronchial epithelial cells of the BEAS-2B cell line were maintained in culture flasks (TPP, Trasadingen, Switzerland) with an F-12 culture medium (Thermo Fisher, Villebon-sur-Yvette, France) supplemented with 10% fetal calf serum (Eurobio, Les Ulis, France), 1% penicillin and streptomycin (Thermo Fisher) and 10 mM HEPES. On day 1, 96-well plates (TPP) were seeded at 1.5 × 10^4^ cells/well in 200 μL of F-12 complemented medium. On Day 0, cells were stimulated or not (control) with 5 μg/mL ultrapure *P. aeruginosa* flagellin (Flagellin) (InvivoGen, San Diego, CA, USA). On Day 2, the cells were then washed twice with fresh medium and incubated for a resting period of 4 days. On Day 6, cells reaching confluence were infected or not (control) with spores of the different species of filamentous fungi or yeast blastospores. Uninfected cells (stimulated with flagellin or not) were incubated in a sterile culture medium for the same time. After 15 h of infection, culture supernatants were collected to determine the antifungal activity, cell viability and inflammatory response of BEC.

### 2.3. Antifungal Activity

To investigate the antifungal activity of BEC, BEAS-2B cells (5.10^4^ cells/wells in 300 µL) were cultured to confluence in chambered coverslips (IBIDI^®^) and then, infected with spores of different species of filamentous fungi or yeast blastospores (10^3^/wells in 300 µL). BEC were infected for 15 h, as at this time, all the fungi or yeast incubated under the same conditions without epithelial cells produced filaments or pseudohyphae, respectively. Fungal growth in the absence of cells served as a control.

Representative pictures of filament formation in the presence or absence of BEC were taken to estimate the fungal growth (EVOS Cell Imaging Systems, San Diego, CA, USA). In addition, supernatants were collected to assay polysaccharides released by fungal strains during growth. Galactomannan, used as an index of *Aspergillus filament* formation, was quantified using the Platelia *Aspergillus* EIA commercial kit (Bio–Rad). Supernatants were treated as sera according to the manufacturer’s recommendations. Mannan, a cell wall polysaccharide released by *Candida* species under pathogenic conditions, was measured using the *Candida* Ag Plus kit (Bio–Rad) according to the recommendations of the manufacturer. β-D-glucan is another cell wall polysaccharide produced by a wide variety of pathogenic fungi, mainly ascomycetes, such as *S. apiospermum* [29]. In preliminary experiments, we observed a correlation between the growth of *S. apiospermum*, estimated under the microscope, and the β-D-glucan concentration detected in the culture supernatant (data not shown). Therefore, we used dosing β-D-glucan, using the FungiTell test (Cape Cod, East Falmouth, MA, USA) following the supplier’s recommendations, to estimate the *S. apiospermum* growth in our experimental conditions.

In all the cases, the antifungal activity was expressed as the percentage of the biomarker values obtained in the presence of BEC divided by the control (growth in the absence of BEC).

### 2.4. Inflammatory Response

The inflammatory response of BEC was evaluated by measuring IL-8 and IL-6, in the supernatants of infection (R&D System, Minneapolis, MN, USA) according to the manufacturer’s recommendations. The impact of flagellin pre-exposure on the inflammatory response of BEC to fungal infections was measured by calculating the following index: cytokine concentrations obtained with non-prestimulated non-infected cells (basal level) subtracted from the values obtained with non-prestimulated and infected cells (response to infection) [27].

### 2.5. Cytotoxicity

The cytotoxicity of fungal infection on BEC was evaluated by measuring the activity of lactate dehydrogenase (LDH) according to the manufacturer recommendations (CytoTox 96 Non-Radioactive kit, Promega, Madison, WI, USA). Data are expressed as a lysis percentage by calculating the ratio of LDH activity measured in the cellular supernatant to the total LDH activity (released LDH plus intracellular LDH) from the same well.

### 2.6. Statistical Analysis

Data are presented as the mean ± SEM. Differences among groups were assessed using Prism version 7.00 (GraphPad Software, San Diego, CA, USA). Mann–Whitney and analysis of variance (ANOVA) with the Bonferroni correction tests were used when two or more quantitative variables were compared, respectively. A difference of *p* < 0.05 was considered statistically significant.

## 3. Results

### 3.1. Inflammatory and Antifungal Response of BEC to Fungal Infections

Respective multiplicity of infection (MOI) values at 2, 0.05, 1, 0.5 and 0.1 were selected for *A. fumigatus*, *A. flavus*, *A. terreus, S. apiospermum* and *C. albicans*, in order to limit BEC cytotoxicity below 10% (Figure 1).

BEC were then infected for 15 h with the defined MOI. Inflammatory response was quantified by measuring IL-8 synthesis, a chemokine involved in the recruitment of neutrophils. All fungal species tested strongly stimulated IL-8 production by BEC (Figure 2B). Interestingly, IL-8 synthesis release by BEC was not related to MOI as *A. flavus* used at the lowest MOI (0.05) induced a higher level of IL-8 release than *S. apiospermum* and *C. albicans* used at MOI of 0.5 and 0.1, respectively.

The antifungal activity of BEC towards each fungal species was then analyzed. As previously demonstrated with *A. fumigatus* [20], we observed in the present study, a reduced formation of hyphae of *Aspergillus* spp. and pseudohyphae of *C. albicans*, in the presence of BEC compared to fungal culture in the absence of cells (Figure 3A–C,E). In contrast, *S. apiospermum* filament formation was not affected by the presence of BEC (Figure 3D).

These microscopic observations were correlated with the reduction in cell wall polysaccharides release in the supernatant during fungal growth (Figure 4). Galactomannan release, used as a surrogate marker of *Aspergillus* growth, was significantly reduced by 43%, 59% and 70% for *A. fumigatus*, *A. flavus* and *A. terreus*, respectively, when fungi were in contact with BEC as compared to the fungal growth in the absence of BEC (Figure 4A–C). In accordance with microscopic observations, the β-D-glucan concentration in the supernatant was similar whether *S. apiospermum* was growing alone or in the presence of BEC (Figure 4D). Although we observed a decrease of 47% in mannan release when *C. albicans* was in contact with BEC, this trend did not reach a significant level (*p* = 0.14) (Figure 4E).

### 3.2. Impact of IIM on the BEC Responses toward Fungi

We investigated the impact of pre-exposure of BEC to *P. aeruginosa* flagellin 4 days before the infection with the different fungal species. To verify there was no bias in our results due to exposure of BEC to flagellin, we first measured the total protein of BEC in each condition to ensure that flagellin did not modify their proliferation capability (data not shown). Then, by measuring the LDH index, we investigated the cytotoxicity of fungi on BEC pre-exposed to flagellin. Regardless of the strain tested, no significant modification in cytotoxicity was observed towards flagellin pre-stimulated cells (data not shown).

We then investigated the impact of flagellin pre-exposition on the ability of BEC to inhibit fungal growth by measuring fungal cell wall polysaccharides in the supernatants. The release of galactomannan was similar whether *A. fumigatus* and *A. flavus* were in contact with BEC pre-stimulated or not with flagellin (Figure 4A,B). In contrast, we observed a significant increase (+14%) in galactomannan release when BEC were pre-stimulated with flagellin and then infected with *A. terreus* (Figure 4C). Finally, pre-stimulation with flagellin did not modify the amount of mannan and β-D-glucan released after *C. albicans* and *S. apiospermum* infection, respectively (Figure 4D,E).

Lastly, we measured the impact of flagellin pre-exposure on the inflammatory response of BEC after 15 h of fungal infection by quantifying IL-8 and IL-6 in cell supernatants (Figure 5). In accordance with our previous study, pre-stimulation with flagellin led to a significant increase in the inflammatory response (1.8-fold for IL-8 and IL-6) of the cells infected with clinical strains of *A. fumigatus* (Figure 5A). A similar trend was observed with *A. flavus* infection for IL-8 (1.37-fold) but not for IL-6 synthesis (Figure 5B). Finally, cytokine synthesis was not modified by flagellin pre-stimulation of cells when they were further infected with *A. terreus* (Figure 5C). On the other hand, BEC pre-stimulated with flagellin and then infected with *S. apiospermum* showed a significant increase in IL-6 synthesis (1.47-fold) but not IL-8 (Figure 5D). Finally, prior contact with *P. aeruginosa* flagellin significantly increased the IL-8 synthesis (1.34-fold) by BEC further infected with *C. albicans*, but not the IL-6 synthesis (Figure 5E).

## 4. Discussion

Chronic lung diseases such as CF or COPD are engaged in a vicious circle where chronic colonization and acute infection favored by altered mucociliary clearance induce chronic inflammation and lung damage, which themselves subsequently favor infections with multiple microbial pathogens [30,31]. By the end, there is a progressive decrease in lung performance, limiting the quality of life of patients, which can lead to end-stage respiratory failure.

BEC play a central role in the pathophysiology of this pathological process. They exhibit antimicrobial activity both through mechanical action (ciliary beating and mucus production) and through the production of soluble mediators such as proinflammatory cytokines and antimicrobial peptides [18,32]. In this study, we deciphered the interactions between BEC and fungal pathogens frequently isolated from the airways of patients suffering from chronic pulmonary diseases. We found that all the tested species (*A. fumigatus, A. flavus, A. terreus, S. apiospermum* and *C. albicans*) are able to trigger an inflammatory response in BEC. The level of the inflammatory response cannot be compared between species because different MOI were used to limit the cytotoxicity of fungi against BEC. As previously described for a lab strain of *A. fumigatus,* we found that BEC exert an antifungal activity against the pathogens tested, with the exception of *S. apiospermum,* that limits their growing capacity. The mechanisms involved in this phenomenon are not fully understood, but the *A. fumigatus* lectin FleA is required to trigger this antifungal activity of BEC toward this species, suggesting a specific PAMP/PRR interaction [20]. It has been described that *S. apiospermum* expresses species-specific cell wall components, such as peptidorhamnomannans [33] or the SapL1 lectin [34] that interact with host cells. Such differences in the cell wall composition of the pathogens could explain the triggering or not of the BEC antifungal activity. Up to today, BEC receptors involved in these interactions are not described.

We recently showed that IIM of BEC induced by exposure to *P. aeruginosa* flagellin led to an exacerbation of the inflammatory response to a subsequent *A. fumigatus* infection with a lab strain [27]. Here, we demonstrated that IIM also influences the responses of pre-stimulated BEC to infection with several other fungal species (Table 1). We confirmed a training effect, with IL-8 and IL-6 synthesis increased, when BEC were further infected with clinical strains of *A. fumigatus*. However, this phenomenon was not restricted to *A. fumigatus*, as the inflammatory response to *S. apiospermum* and *C. albicans* infection was also increased after flagellin pre-exposure. This result is of major importance, as the occurrence of concomitant colonization with *P. aeruginosa* and the fungal pathogens tested in this study is commonly encountered in patients suffering from chronic pulmonary diseases, notably CF [35,36]. Nonetheless, the training effect was observed only for IL-6 or for IL-8 when BEC were further infected with *S. apiospermum* or *C. albicans,* respectively. Thus, IL-6 and IL-8 synthesis of BEC pre-exposed to flagellin was not affected in the same way depending on the fungal species involved. Based on what has been shown for macrophages, one can speculate that flagellin can modulate epigenetic processes, leading to the activation and/or inhibition of distinct pathways involved in the synthesis of IL-8 and IL-6, induced by the fungal species encountered. Moreover, IL-6 synthesis in BEC infected by *C. albicans* after their pre-exposition to flagellin showed a trend toward tolerance. Interestingly, BEC responses to *C. albicans* can be compared to data previously obtained in a mouse model of keratitis [37]. In this case, mice received the topical administration of flagellin into the cornea, 6 to 72 h before infection with *C. albicans*. The authors showed that pre-exposure to flagellin increased the clearance of the fungal pathogen at the first stages of infection by a neutrophil-dependent mechanism, but also that flagellin reduced inflammation in response to the fungus infection responsible for the damaged corneas. Our results support these observations. Indeed, in our model with *C. albicans*, we observed that the pre-stimulation of BEC by flagellin induced, on the one hand, training for the synthesis of IL-8, a chemokine which ensures the recruitment of neutrophils to the site of infection, and, on the other hand, a trend to a tolerant response for IL-6, thus reducing the inflammatory environment, which possibly limits cellular damage.

In contrast, flagellin pre-stimulation did not modify the antifungal activity of BEC towards *A. fumigatus, A. flavus* and *C. albicans*, suggesting that signaling pathways involved in this process are not modulated by flagellin exposure. However, we noticed a reduced antifungal activity of pre-stimulated BEC against *A. terreus,* based on increased galactomannan release in the cell supernatant after flagellin pre-stimulation, which has not been observed microscopically. We can hypothesize that there is some down-regulation of specific PRR that limits recognition of the fungal pathogen as it has been shown for TLR4, a receptor of bacterial lipopolysaccharide [18].

## 5. Conclusions

In conclusion, we have demonstrated in this study that BEC, in addition, to being a keystone of the mucociliary escalator, are able to recognize inhaled fungal pathogens, to mount an immune response and to limit fungal growth. Some of these responses are modulated after the pre-exposure of BEC to *P. aeruginosa* flagellin. We thus hypothesize that IIM established during the exposure of BEC to bacterial infections could influence the pathophysiology of subsequent infections with other fungal pathogens. This is to be compared to the observation that the Bacille Calmette Guerin vaccine significantly reduce the severity of SARS-CoV2-associated symptoms [38]. This suggests that an unspecific stimulation could be used to reinforce the immune system towards further infection. In addition, considering the major role of inflammation in the pathophysiology of chronic lung diseases such as CF or COPD, understanding the intracellular processes induced by bacterial colonization could open new therapeutic opportunities in the development of preventive and antifungal treatments.

## Figures and Tables

**Figure 1 jof-08-01268-f001:**
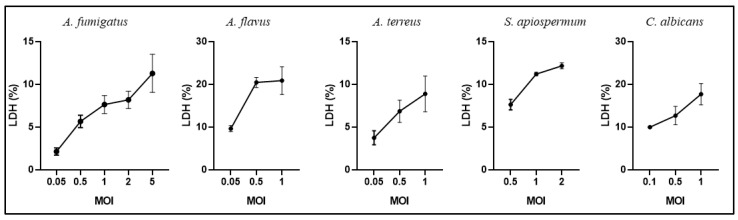
Cytotoxicity of fungal species as measured with LDH assay according to the MOI used, following 15 h of infection with each strain of the five species. The data were expressed as a lysis percentage by calculating the ratio of LDH activity measured in the cellular supernatant to total LDH activity (released LDH plus intracellular LDH = 100%) from the same well. Each point represents the mean ± standard error of the mean (SEM) of triplicate of one experiment with each strain.

**Figure 2 jof-08-01268-f002:**
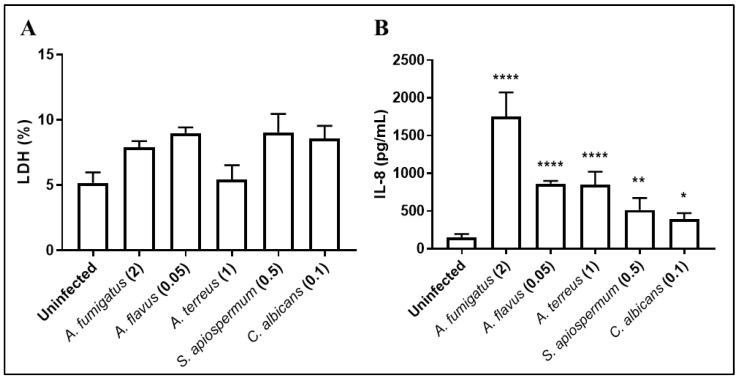
Cytotoxicity of *A. fumigatus*, *A. flavus*, *A. terreus*, *S. apiospermum* and *C. albicans*, and inflammatory response of BEC, following 15 h of infection (**A**). IL-8 production is expressed as pg/mL. Numbers in parentheses indicate MOI used for cell infections (**B**). Each histogram is compared with uninfected group and represents the mean ± SEM of 3 independent experiments, * *p* < 0.05; ** *p* < 0.01; **** *p* < 0.0001 (ANOVA Bonferroni).

**Figure 3 jof-08-01268-f003:**
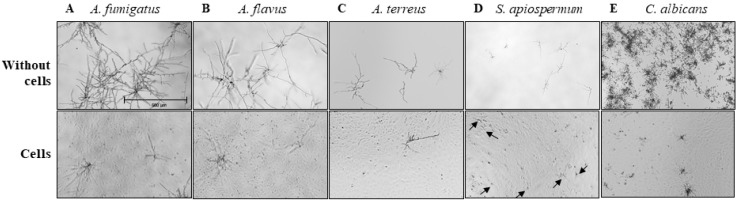
Microscopic observations of fungal growth by clinical isolates of *A. fumigatus* (**A**), *A. flavus* (**B**), *A. terreus* (**C**), *S. apiospermum* (**D**) and *C. albicans* (**E**) in the absence (**top**) or presence (**bottom**) of BEC after 15 h of infection (magnification, 40×). Black arrows indicate filaments of *S. apiospermum* cultured with BEC (**D**).

**Figure 4 jof-08-01268-f004:**
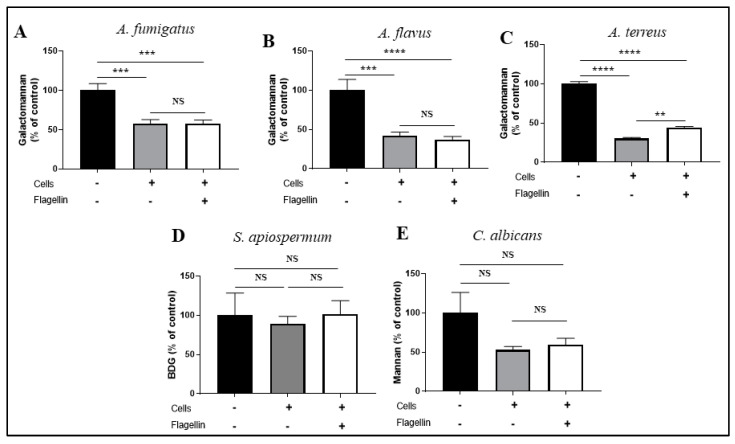
Fungal growth measurement after 15 h of infection under 3 conditions: without BEC representing growth control (100%) (black bars), with BEC (gray bars) and with *P. aeruginosa* flagellin pre-stimulated BEC (white bars). Fungal growth was quantified in the culture supernatant by using a galactomannan assay for *Aspergillus* species: 4 strains of *A. fumigatus* (**A**), 4 strains of *A. flavus* (**B**) and 2 strains of *A. terreus* (**C**); ß-D-glucan assay for the 2 clinical strains of *S. apiospermum* (**D**) and a mannan assay for the 3 clinical strains of *C. albicans* (**E**). Each bar represents the mean ± SEM of 3 independent experiments, BDG: β-D-glucan; NS: nonsignificant; ** *p* < 0.01; *** *p* < 0.001; **** *p* < 0.0001 (ANOVA Bonferroni).

**Figure 5 jof-08-01268-f005:**
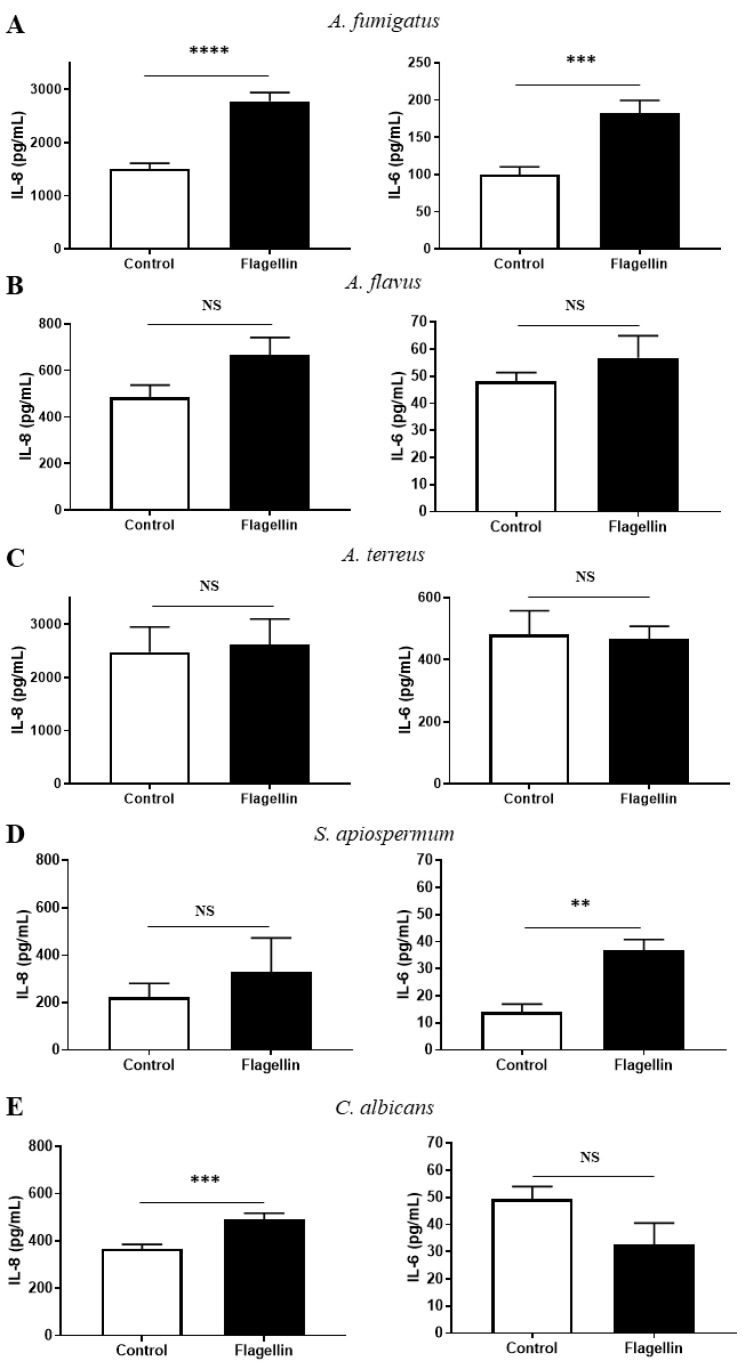
Inflammatory response of BEC pre-stimulated (flagellin, black bars) or not (control, white bars) with *P. aeruginosa* flagellin and then infected for 15 h with 4 strains of *A. fumigatus* (**A**), 4 strains of *A. flavus* (**B**), 2 strains of *A. terreus* (**C**), 2 strains of *S. apiospermum* (**D**) and 3 strains of *C. albicans* (**E**). IL-8 and IL-6 production are expressed as pg/mL after subtraction of the values obtained for non-infected conditions. Each histogram represents the mean ± SEM of 3 independent experiments, NS: non-significant; ** *p* < 0.01; *** *p* < 0.001; **** *p* < 0.0001 (Mann–Whitney test).

**Table 1 jof-08-01268-t001:** Impact of *P. aeruginosa* flagellin pre-exposure on the inflammatory and antifungal responses of BEC subsequently infected with different fungal pathogens. → no impact; ↑ increased response; ↓ decreased response.

	*A. fumigatus*	*A. flavus*	*A. terreus*	*S. apiospermum*	*C. albicans*
**Cytotoxicity**	→	→	→	→	→
**Antifungal Activity**	→	→	↓	→	→
**IL-8 Synthesis**	↑	↑	→	→	↑
**IL-6 Synthesis**	↑	→	→	↑	→

## Data Availability

Not applicable.

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
