# Peer review of "Effect of Flagellin Pre-Exposure on the Inflammatory and Antifungal Response of Bronchial Epithelial Cells to Fungal Pathogens"

_jof, 2022, doi:10.3390/jof8121268_

Round 1

Reviewer 1 Report

The manuscript entitled "Effect of flagellin pre-exposure on the inflammatory and antifungal response of bronchial epithelial cells to fungal pathogens" by Bigot and co-authors report the investigation of the roles played by innate immune memory of bronchial epithelial cells, achieved by in pre-exposure to  Pseudomonas aeruginosa flagellin, on the responses to infection by Aspergillus spp., Scedosporium apiospermum and Candida albicans. The manuscript is well organized, writing is easy to follow and the conclusions are supported by the results. This represents a nice piece of work, well organized and on a topic of relevance when considering respiratory infections as is the case of cystic fibrosis patients. As highlighted by the authors, in many cases these patients are colonized initially with Pseudomonas aeruginosa, and at advanced stages, with Aspergilus fumigatus, and often its clinical significance remains unclear. There two minor issues that the authors should address, which should not be an obstacle to the acceptance of the manuscript: 

line 25, substitute "others" by "other"

Figure 1 legend: points are represented as circles, squares and triangles. No explanation is given for this. Is there a reason for this distinct representation?

Author Response

We thank the reviewer for his comments. The English modifications have been made.
Concerning figure 1, the shapes of the dots had no meaning and we have homogenized the representation by putting only circles.

Reviewer 2 Report

The report is devoted to the study of the protective properties of bronchial epithelial cells (BEC) against various strains of 3 genera of fungi often associated with bacterial lung infections. The effect of BEC pretreatment with P. aeruginosa flagellin on the possibility of enhancing the protective properties of BEC was also investigated. The results of the study may serve as a prologue to further understanding of the search for treatment of severe chronic pulmonary diseases.

The methodology used in the study and all laboratory procedures are thoroughly described in the Methods section, sufficient to reproduce the study.

Question-1: What does 100% correspond to in each panel in Figure 1? This should be stated in the legend of the figure. Is "100%" the same in all five panels?

The text is written in good English, sufficient to understand the study. However, some sentences should be corrected throughout the text as shown below, taking into consideration the following notations: [...] for inclusion and >...< for deletion:

Abstract

25:  to >others<  [other] fungal infections

30:  >Deepen< [deepening] our knowledge

Introduction

65: The [bronchial] immune defense >at the bronchial<

67: >either<   secrete

68: ciliary  >beat<    [beating]  of BEC

Methods:

95: >representative of<  [representating] the most ...

116: described >in<  [by]

163: >Cytotoxixity<  [Cytotoxicity]

Results

186: >the production IL-8<  [IL-8 production]

207: observations   >were<   correlated

228-230: To verify >that<  [there was] no bias  >could be induce<    in our results [due to exposure of BEC to flagellin]    >by flagellin exposition of BEC<   , we first measured total protein of BEC in each condition to ensure that flagellin did not modify their proliferation capability.

239: flagellin  [and]  then

Discussion

277: BEC exert  [an antifungal activity]  against

278: >an antifungal activity<    that limits   

282: S. apiospermum  >express<   [expresses]

303: synthesis   >of<   [in]  BEC

310: to the fungus  >, inflammation<   [infection]  responsible

312: [,] on the one hand [,]

314: [,] on the other hand [,] a trend to   >tolerance<    [a tolerant]  response for IL-6, >allowing for a less<  [thus reducing the] inflammatory environment [, which]  >that could<     possibly   >limit<   [limits] cellular damage.

321-322: [that]  there is some down-regulation of specific PRR   >limiting<   [that limits] recognition

324: Impact of P. aeruginosa [flagellin] pre-exposure

Author Response

We thank the reviewer for his comments.

The English modifications have been made.

Concerning the legend of figure 1, we have added this sentence to explain the calculation "The data were expressed as a lysis percentage by calculating the ratio of LDH activity measured in the cellular supernatant to total LDH activity (released LDH plus intracellular LDH = 100%) from the same well."